# Influence of Inflammatory Pain and Dopamine on Synaptic Transmission in the Mouse ACC

**DOI:** 10.3390/ijms241311113

**Published:** 2023-07-05

**Authors:** Soroush Darvish-Ghane, Jennet Baumbach, Loren J. Martin

**Affiliations:** 1Department of Cell and Systems Biology, University of Toronto, Toronto, ON M5S 3G5, Canada; soroush.darvish.ghane@mail.utoronto.ca; 2Department of Psychology, University of Toronto Mississauga, Mississauga, ON L5L 1C6, Canada; jennet.baumbach@mail.utoronto.ca

**Keywords:** ACC, excitatory, inhibitory, dopamine, inflammation, pain, GABA_A_R, AMPAR

## Abstract

Dopamine (DA) inhibits excitatory synaptic transmission in the anterior cingulate cortex (ACC), a brain region involved in the sensory and affective processing of pain. However, the DA modulation of inhibitory synaptic transmission in the ACC and its alteration of the excitatory/inhibitory (E/I) balance remains relatively understudied. Using patch-clamp recordings, we demonstrate that neither DA applied directly to the tissue slice nor complete Freund’s adjuvant (CFA) injected into the hind paw significantly impacted excitatory currents (eEPSCs) in the ACC, when recorded without pharmacological isolation. However, individual neurons exhibited varied responses to DA, with some showing inhibition, potentiation, or no response. The degree of eEPSC inhibition by DA was higher in naïve slices compared to that in the CFA condition. The baseline inhibitory currents (eIPSCs) were greater in the CFA-treated slices, and DA specifically inhibited eIPSCs in the CFA-treated, but not naïve group. DA and CFA treatment did not alter the balance between excitatory and inhibitory currents. Spontaneous synaptic activity revealed that DA reduced the frequency of the excitatory currents in CFA-treated mice and decreased the amplitude of the inhibitory currents, specifically in CFA-treated mice. However, the overall synaptic drive remained similar between the naïve and CFA-treated mice. Additionally, GABAergic currents were pharmacologically isolated and found to be robustly inhibited by DA through postsynaptic D2 receptors and G-protein activity. Overall, the study suggests that CFA-induced inflammation and DA do not significantly affect the balance between excitatory and inhibitory currents in ACC neurons, but activity-dependent changes may be observed in the DA modulation of presynaptic glutamate release in the presence of inflammation.

## 1. Introduction

Chronic pain is an unmet medical need affecting approximately 20% of the population worldwide [1,2,3,4]. Despite the prevalence of chronic pain syndromes, current treatment options are often ineffective, expensive, or have addictive potential [5,6]. To date, pain research at the preclinical level has sought to understand chronic pain’s nociceptive and spinal contributors, with a poorer mechanistic understanding of cortical brain involvement [1,7]. However, understanding the dysregulation of cortical brain regions and circuits may be critical to improving treatment options for chronic pain sufferers. Evidence from rodent and human studies shows that the anterior cingulate cortex (ACC) is involved in acute and chronic pain [8,9,10,11]. Some work shows that synaptic and microcircuit connectivity in the ACC is altered in chronic pain models [12]. Still, it remains unknown how the balance of excitatory and inhibitory synaptic transmission in the ACC is altered in chronic pain states. 

Several studies have provided evidence that multiple pain models involve alterations in the balance of excitatory/inhibitory (E/I) neurotransmission in the medial prefrontal cortex (mPFC) and somatosensory regions [13,14]. Synaptic E/I balance is vital for network stability and information processing [15]; however, the relative role of inhibitory transmission in the ACC during pain states remains largely unexplored. In human osteoarthritis patients, pain intensity negatively correlates with GABA levels in the ACC [16]. In mice, sciatic nerve injury, a model of neuropathic pain, reduces the connectivity between excitatory and inhibitory ACC neurons [17]. There is also evidence that enhanced GABAergic transmission in the ACC is associated with antinociceptive function [18,19], and mice with chronic inflammation have depressed presynaptic GABAergic transmission in the ACC [20]. Furthermore, transplanting interneurons within the ACC restores GABAergic tone and reduces pain-related aversion in mice with chronic pain [19]. Hence, pain processing appears to be inversely related to GABAergic transmission in the ACC.

There is ample evidence for the involvement of central dopaminergic circuits in chronic pain states [21,22,23,24]. Animal studies indicate a hypodopaminergic tone in chronic pain [22,25,26], such that pain chronification can be viewed as a state of reward deficiency [27]. In vivo infusion of DA to the ACC has been shown to have long-term antinociceptive properties [28], and DA receptor activation in ACC brain slices inhibits glutamatergic excitatory AMPAR transmission [29,30,31]. Furthermore, dopamine receptor type 1 (D1R) activation reduces the excitability of ACC neurons, and microinjection of a D1R agonist in the ACC relieves pain in mice with nerve injury [32]. Exogenous DA application to ACC brain slices reduces evoked excitatory postsynaptic currents [30]. The application of D1R agonists mimics this effect [29]. Given that DA modulates the balance of E/I transmission in mPFC neurons to alter neuronal output [33], we sought to investigate the dopaminergic modulation of E/I transmission in the ACC of mice with and without inflammatory pain. 

Here, we aimed to investigate the effects of DA and complete Freund’s adjuvant (CFA)−induced inflammation on excitatory and inhibitory currents in ACC neurons. CFA is composed of a mixture of mineral oil, typically paraffin oil, and heat-killed Mycobacterium tuberculosis bacteria. The heat-killed bacteria serve as an immune system activator, stimulating a strong immune response when combined with an antigen that results in enhanced nociceptive behavior when administered to rodents. In the present study, most electrophysiological recordings were conducted without pharmacological isolation of the currents, allowing for a comprehensive examination of their modulation. Instead of pharmacological isolation, most currents were recorded by maintaining neurons at the reversal potential for the opposite current. In this study, these currents are referred to as eEPSCs^clamped^ or eIPSCs^clamped^, distinguishing them from eEPSCs or eIPSCs that have been pharmacologically isolated, which are denoted as eEPSCs^pharm^ or eIPSCs^pharm^ (i.e., with glutamate or GABA receptor blockers in the slice bath). We found that neither DA nor CFA treatment significantly altered eEPSCs^clamped^ in the ACC. However, when analyzing individual neurons, some eEPSCs^clamped^ exhibited inhibitory responses, while others showed potentiation or were unresponsive to DA application. The degree of eEPSC^clamped^ inhibition was significantly higher in naïve slices compared to that in the CFA condition. Conversely, baseline eIPSCs^clamped^ were greater in slices from CFA-treated mice and DA inhibited eIPSCs^clamped^, particularly in the CFA-treated group. The overall E/I balance did not differ between naïve and CFA-treated mice and was not altered by slice application of DA in either group. We also explored spontaneous excitatory and inhibitory currents (sEPSCs and sIPSCs) as a measure of synaptic function and found that DA reduced the frequency of sEPSCs and decreased the amplitude of sIPSCs in CFA-treated mice. However, synaptic drive remained similar between naïve and CFA-treated mice. Furthermore, DA robustly inhibited eIPSCs^pharm^ in naïve mice. Postsynaptic D2 receptors and G-protein activity mediated this inhibition. These findings indicate that CFA-induced inflammation and DA do not significantly alter the E/I balance in ACC neurons. However, activity-dependent changes were observed in the DA modulation of presynaptic glutamate release in the presence of inflammation and pharmacological isolation of eIPSCs.

## 2. Results

### 2.1. DA and CFA-Induced Inflammation Do Not Alter E/I Balance in the ACC

Based on our previous work, we sought to comprehensively examine the DA modulation of excitatory and inhibitory currents within individual ACC neurons following inflammatory injury. However, unlike our previous studies, eEPSCs and eIPSCs were recorded in the same neuron in drug-free aCSF without pharmacological isolation. Instead, the EPSC and IPSC components were isolated by clamping the cells at inhibitory (−60 mV) and excitatory (0 mV) reversal potentials, respectively. This is a common approach when recording E/I currents in the same neuron, as pharmacological blockers (e.g., CNQX, APV, CGP-55845) may alter postsynaptic potentials following washout [34]. EPSCs and IPSCs, recorded without any blockers in the bath, are herein referred to as evoked clamped EPSCs (eEPSC^clamped^) and IPSCs (eIPSC^clamped^).

We obtained 10–15 min of stable baseline currents for eEPSCs^clamped^ (V_h_ = −60 mV) and eIPSCs^clamped^ (V_h_ = 0 mV), before and after the bath application of DA. In ACC neurons, eEPSCs^clamped^ did not differ between naïve and CFA-treated mice, and DA did not significantly inhibit eEPSCs^clamped^ in either condition (Figure 1a). This was surprising, given that we previously showed DA inhibition of eEPSCs in the ACC of slices from naïve and CFA-treated mice [31]. However, a primary difference between the current work and our previous studies is that our prior work recorded eEPSCs in the presence of pharmacological blockers (e.g., picrotoxin, CGP-55845, APV). We noted that following DA application, eEPSCs^clamped^ were either inhibited (naïve: 9/15 of eEPSCs; CFA: 4/9 of eEPSCs) or potentiated (naïve: 4/15 of eEPSCs; CFA: 4/9 of eEPSCs). Some neurons in both groups were unresponsive to DA application (naïve: 2/15 of eEPSCs; CFA: 1/9 of eEPSCs). Thus, neurons were separated and analyzed based on whether they showed an overall inhibitory or potentiated response. Statistical comparison between the degree of eEPSC^clamped^ inhibition or potentiation as the percentage change of the baseline revealed a significantly higher level of inhibition of eEPSCs in naive slices relative to the CFA condition, which aligns with our previous findings [31]. At the same time, no differences emerged between potentiated eEPSCs (Figure 1b,c). 

In addition, we recorded eIPSCs to understand whether CFA treatment or DA application altered inhibitory currents. Overall, baseline eIPSCs^clamped^ were larger in the CFA group, while DA inhibited eIPSCs^clamped^ in CFA, but not naïve slices (Figure 1d). Unlike eEPSCs^clamped^, most of the eIPSCs^clamped^ from naïve and CFA mice showed an overall inhibitory response (Figure 1e,f). In naïve and CFA slices, the majority (naïve: 10/15 of eIPSCs; CFA: 6/9 of eIPSCs) showed an overall inhibitory response to DA application, while neurons showing potentiated eIPSC^clamped^ made up a small proportion in both groups (naïve: 2/15 of eIPSCs; CFA: 1/9 of eIPSCs). Some neurons in both groups were unresponsive to DA application (naïve: 3/15 of eIPSCs; CFA: 2/9 of eIPSCs). Thus, we analyzed the degree of inhibition in neurons based on whether they showed an overall inhibitory response. Given the small sample size, we did not consider potentiated or unresponsive neurons. The percentage of baseline inhibition of eIPSCs^clamped^ was similar between naïve and CFA-treated slices (Figure 1f). The percentage change of eIPSCs^clamped^ in this experiment was approximately 20% in both naïve and CFA-treated slices. 

The overall E/I ratio between naïve and CFA-treated mice was not different or altered by DA in either group (Figure 2a). To further understand whether DA disproportionally influenced the relationship between eEPSCs^clamped^ and eIPSCs^clamped^ within individual neurons, eEPSCs^clamped^ and eIPSCs^clamped^ amplitudes were plotted against each other. A linear relationship between eEPSCs^clamped^ and eIPSCs^clamped^ in ACC neurons was found for naïve and CFA-treated mice, before and after DA application (Figure 2b,c). The regression slopes did not differ for naïve and CFA-treated mice following DA application, suggesting that CFA-induced inflammation did not alter the E/I balance of DA in the ACC neurons (Figure 2b,c). 

### 2.2. DA Modulates Spontaneous Excitatory and Inhibitory Currents in the ACC

In addition to evoked ionotropic excitatory and inhibitory transmission, we investigated spontaneous EPSCs (sEPSC) and IPSCS (sIPSCs) as a global read-out of synaptic function in ACC neurons. These recordings were conducted in blocker-free aCSF, as performed in our previous studies [35]. In CFA-treated mice, DA application significantly reduced the frequency of sEPSCs, whereas no change was evident in naïve mice (Figure 3a,b). The amplitude of sEPSCs was similar between naïve and CFA-treated mice, before and after DA application (Figure 3a,c). There was an overall increase in the frequency of sIPSCs in CFA-treated mice; however, DA did not alter sIPSC frequency in naïve or CFA-treated mice (Figure 3d,e). In CFA-treated mice, DA application significantly reduced the amplitude of sIPSCs, whereas no change was observed in naïve mice (Figure 3d,f). The overall synaptic function was not altered between naïve and CFA-treated mice, as the synaptic drive was similar between the groups (Figure 3g). However, a significant drug effect emerged, with synaptic drive shifted towards inhibition, only in CFA-treated mice following DA application (Figure 3d,g). These results demonstrate that CFA-induced inflammation caused activity-dependent changes in the DA modulation of ACC presynaptic glutamate release.

### 2.3. DA inhibits Pharmacologically Isolated Evoked GABAergic Currents in ACC of Mice

Previous work demonstrated that activation of DA receptors in ACC brain slices inhibits pharmacologically isolated AMPAR-mediated EPSCs (eEPSC) in the ACC of naïve mice [29,30,31]. However, this was not true for evoked EPSCs recorded without pharmacological isolation (i.e., eEPSC^clamped^). Given that we previously showed DA inhibits pharmacologically isolated eEPSCs in an earlier study [31], we sought to pharmacologically isolate GABAergic currents in the ACC and determine whether DA modulates these responses when pharmacologically isolated. Thus, we recorded GABA_A_R-mediated eIPSCs (V_h_ = 0 mV) that were pharmacologically isolated by including CNQX, APV, and CGP-55845 in the slice bath to block AMPA, NMDA, and GABA_B_ receptors, respectively. Pharmacologically isolated eIPSCs, referred to as eIPSC^pharm^, were blocked by the application of PTX (100 μM), a GABA_A_R blocker indicating that GABA_A_Rs mediated this current. The transient application of DA (50 μM) for 10 min significantly inhibited GABAergic eIPSCs^pham^ in the ACC of naïve mice (Figure 4a). Subsequently, washing out DA from the bath reversed this inhibition (Figure 4b). DA at 20 μM and 100 μM also inhibited eIPSCs, but this inhibition was not significantly different from that of the 50 μM application of DA (Figure 4c). 

To test whether the inhibitory effect of DA on eIPSCs^pharm^ occurred via a pre- or postsynaptic mechanism, we measured the paired–pulse ratio of eIPSCs^pharm^ at baseline and following DA application (Figure 5a–c). DA did not induce a significant change in the PPR of eIPSCs^pharm^ (paired *t*-test, *t*_7_ = 0.6, *p* = 0.55; Figure 5b,c). However, postsynaptic inhibition of G-protein activity by guanosine 5′-[beta-thio] diphosphate (GDP-β-S; 2 mM) occluded the inhibition of eIPSCs^pharm^ by DA (Figure 5d–f). Since sulpiride (100 μM), a D2R antagonist, blocks the DA-mediated inhibition of pharmacologically isolated eEPSCs [30], we tested whether inhibiting eIPSCs in the ACC operated via a similar mechanism. DA no longer inhibited eIPSCs^pharm^ when co-applied with sulpiride (Figure 5g–i). These results indicate that postsynaptic D2 receptors, coupled with GPCR activation, induce the inhibition of eIPSCs^pharm^ in the ACC, in a mechanism similar to the DA-induced inhibition of eEPSCs^pharm^ in mouse brain slices [30]. 

## 3. Discussion

Given the evidence that DA signaling in the ACC modulates the sensory and affective processing of pain [28,29,36], we investigated the manner in which DA modulates rapid ionotropic E/I transmission in the ACC. In the present study, we demonstrate that DA does not inhibit eEPSCs, when recorded without pharmacological isolation of the current (i.e., eEPSCs^clamped^). This finding differed from that in our previous work, in which we showed that DA inhibits pharmacologically isolated eEPSCs (i.e., eEPSCs^pharm^) [31]. This result led us to categorize neurons based on whether eEPSCs^clamped^ showed an overall inhibitory or potentiated response following DA application. In neurons in which eEPSCs^clamped^ were inhibited by DA, there was less eEPSCs^clamped^ inhibition in the CFA than in the naïve condition, which reflects our previous work with pharmacologically isolated eEPSCs (i.e., eEPSCs^pharm^) [31]. However, the overall power of these comparisons was low, given that the data were separated data based on whether DA inhibited or potentiated the current (Figure 1c). Thus, even though we report these findings as statistically significant, further validation and confirmation of these observations should be conducted using a larger sample size. This will improve the reliability and robustness of the results as a whole. Conversely, eIPSCs^clamped^ in the ACC neurons of CFA-treated mice showed an overall inhibitory response, while eIPSCs^clamped^ from pain naïve mice were not inhibited by DA. Consistent inhibition by DA was observed when eIPSCs were pharmacologically isolated, as eIPSCs^pharm^ showed robust inhibition by DA in naïve ACC neurons. Moreover, CFA treatment and DA application did not significantly alter E/I balance in ACC neurons. Interestingly, DA significantly reduced the probability of presynaptic glutamate release in CFA-treated mice, which resulted in reduced synaptic drive measured as a function of spontaneous transmission. This was unexpected, as our previous studies indicated that a presynaptic mechanism did not underly eEPSCs^pharm^ inhibition by DA in the ACC [31]. Finally, the bath application of DA transiently inhibited the peak amplitude of eIPSCs^pharm^ in the ACC of mice through postsynaptic GPCR activation dependent on D2Rs. Our results, combined with those of our previous work, indicate that the DA modulation of ACC currents in naïve and CFA-treated mice may be influenced by the recording condition and/or presynaptic receptor mechanisms. 

Previous pharmacological and electrophysiological experiments in the ACC of naïve mice demonstrated that DA reversibly inhibits AMPAR-mediated eEPSCs by a postsynaptic GPCR mechanism involving D1 and D2Rs [30]. In the current experiments, we did not observe a robust inhibition of eEPSCs or eIPSCs, when recorded without pharmacological blockers. This contrasts with the results of our previous work, in which we observed significant inhibition of pharmacologically isolated eEPSCs [31] and eIPSCs, as shown in the current study. The current work, combined with our previous studies [29,31], indicates that differences in ACC neuronal responses to DA are heterogeneous and may be associated with microcircuits or may be susceptible to presynaptic modulation. For instance, presynaptic glutamate receptors regulate the release of neurotransmitters, including GABA, from presynaptic terminals [37,38]. Presynaptic cortical neurons can release glutamate, GABA, or a mixed glutamate/GABA co-release [39]. In neurons receiving mixed glutamate/GABA co-release, the kinetics of postsynaptic currents resemble those observed in neurons receiving only glutamate, but they differ from neurons receiving only GABA [39]. Moreover, DA receptors, including D1-like (D1 and D5) and D2-like (D2, D3, and D4) receptors, are found both pre- and post-synaptically, including on GABAergic neurons where they are known to modulate neurotransmitter release [40,41]. Generally, D1-like receptors tend to produce excitatory effects, while D2-like receptors often show inhibitory effects on GABAergic transmission. Thus, it is also possible that neurons express different complements of DA receptors, which influenced our recordings. However, the effects can be more complex, and the net result depends on the interplay between the specific dopamine receptor subtype, the neuronal circuitry, and the presence of other neurotransmitters and modulators. 

In our recordings, in which glutamate or GABA_A/B_ receptors were not pharmacologically blocked, postsynaptic responses may have been influenced by presynaptic glutamate and GABA receptors and the DA modulation of these responses. Thus, we suspect that the heterogeneity in the postsynaptic modulation of excitatory and inhibitory currents by CFA treatment and DA was unveiled in our current study because most recordings did not block presynaptic (and postsynaptic) receptors that may impact postsynaptic responses. This could indicate that postsynaptic responses may suffer from variability in their activation characteristics and responses to DA, depending on the presynaptic release machinery. Furthermore, a high concentration of the D2R antagonist sulpiride (100μM) blocked the dopaminergic inhibition of eIPSCs^pharm^, similar to the results of a previous report in which sulpiride 100 μM also blocked the DA-mediated inhibition of eEPSCs [30]. The similarity of the kinetics, reversibility of eIPSC inhibition to eEPSC inhibition, and sensitivity of the inhibition to sulpiride suggests that DA rapidly modulates GABAergic and glutamatergic ionic channels through a common postsynaptic mechanism, but only when these currents are pharmacologically isolated. DA is known to modulate the plasticity of AMPAR transmission by inducing changes in the phosphorylation status of the receptors [42]. Due to the rapid and reversible nature of the DA modulation of E/I transmission, a mechanism involving the dynamic phosphorylation events of these rapid inotropic transmissions to tune the synaptic E/I current may exist. As a possible mechanism, future work needs to address the phosphorylation status of AMPA and GABA receptors in response to selective DA receptor activation in genetically defined neuronal populations.

We also investigated spontaneous ionotropic transmission, which can be modulated in two ways. The frequency of events indicates the probability of neurotransmitter release, while amplitude measures the postsynaptic responsivity to neurotransmitter release. In naïve mice, DA application did not significantly affect the frequency or amplitude of spontaneous E/I transmission. However, in CFA-treated mice, DA significantly reduced the frequency of spontaneous glutamatergic transmission, which indicates the probability of a reduction in presynaptic release. Our previous work using the CFA inflammation model demonstrated that DA reduces the probability of evoked glutamatergic transmission [31]. These results collectively indicate that continuous peripheral inflammation may introduce a new mode of synaptic dopaminergic modulation in the ACC. Since potentiation of presynaptic glutamate release probability in the ACC is a possible mechanism for anxiety during pain [43], DA release in the ACC during pain may subserve an anxiolytic function. The possible employment of presynaptic DA receptor recruitment and/or postsynaptic DA-dependent retrograde mechanisms of presynaptic inhibition [44] are potential mechanistic candidates for the DA’s inhibitory function.

Previous preclinical studies regarding CFA-induced inflammatory pain demonstrated the probability of the potentiation of presynaptic glutamate release in the ACC of mice. Our experiments compared evoked E/I currents in naïve mice and those with CFA treatment. We determined to test mice following four days of CFA-induced inflammation, based on the previous finding that this time point coincides with robust hypersensitivity to mechanical stimuli and reduced dopaminergic inhibition of AMPAR currents [31]. Hence, inflammation induces activity-dependent changes in DA receptor signaling and ionotropic transmission in the ACC. In the ACC, postsynaptic depolarization and NMDAR activation can induce potentiation of AMPAR-mediated eEPSCs [8,45], and D1R activation with low agonist concentrations has been shown to stimulate mechanisms of NMDAR-dependent LTP induction in the ACC [46]. Since eIPSCs in these neurons were recorded at a depolarized membrane potential (0 mV) for several minutes, the depolarization-induced activation of NMDAR may contribute to the selective potentiation of eEPSCs in specific ACC neurons. Experiments monitoring DA activity of isolated AMPARs voltage clamped at −60 mV throughout the entire course of the experiment prevent activation of NMDAR mediated mechanisms from inducing potentiation. Further, ACC neurons with excitatory projections to the spinal dorsal horn possess an inhibitory DA signaling pathway, suggesting that endogenous release of DA onto these neurons would reduce the effect on nociceptive transmission [47]. A DA-induced increase in the inhibition of these neurons in naïve mice is consistent with the DA receptor-mediated reduction of mechanical sensitivity in the ACC [29]. However, the potentiation of specific ACC neurons may be involved in modulating mechanical sensitivity, depending on the projection of these neurons. 

Overall, our findings highlight the heterogeneity of ACC neuronal responses to DA, depending on pain state and presynaptic modulation. The implications of our results are twofold. Firstly, they emphasize the importance of considering the specific experimental conditions and the presence of presynaptic receptors when studying the effects of DA on synaptic currents in the ACC. This highlights the need for precise experimental design and methodology to accurately interpret DA modulation in neuronal circuits. Secondly, the results suggest that DA modulation in the ACC may play a role in pain-related mechanisms and potentially contribute to pain modulation. The observed reduction in presynaptic glutamate release in response to DA in CFA-treated mice and the heterogeneity of ACC neuronal responses imply a complex interplay between DA, pain processing, and synaptic transmission. However, it should be noted that we did not confirm CFA-induced mechanical sensitivity in the mice used for electrophysiological recordings, which may have played a role in the overall heterogeneity of the ACC synaptic responses. Future studies are necessary to address the role of ACC DA signaling in pain processing under abnormal DA transmission conditions.

## 4. Material and Methods

### 4.1. Animals

Male adult (4 to 6 weeks of age) C57BL/6J mice were acquired from the Jackson Laboratory (Bar Harbor, ME, USA) and used for all experiments. All mice were housed in groups of 4 upon arrival. Procedures followed the animal care standards set forth by the Canadian Council on Animal Care (CCAC) and approved by the University of Toronto’s Biosciences Panel on Laboratory Animal Care. All animals were maintained within a temperature-controlled environment (20 ± 1° C), with a 12:12 h light:dark cycle. A compressed cotton nesting square and crinkled paper bedding were provided in each cage as a source of environmental enrichment. All mice had access to food (Harlan Teklad 8604) and water ad libitum.

### 4.2. Tissue Preparation for Electrophysiology

Mice were anesthetized with 5% isoflurane and humanely euthanized by decapitation. The brains were quickly removed and placed in cold (4 °C) oxygenated (95% O_2_; 5% CO_2_) artificial cerebrospinal fluid (aCSF) consisting of (in mM) 124 NaCl, 4.4 KCl, 2 CaCl_2_, 1 MgSO_4_, 25 NaHCO_3_, 1 NaH_2_PO_4_, and 10 glucose. Brain slices (300 μm) containing coronal sections of the ACC were prepared with a VT1200S tissue slicer (Leica, Concord, ON, Canada). The slices were allowed to recover for a minimum of 60 min in a submerged holding chamber (25 °C) before recording was conducted. 

### 4.3. Whole-Cell Patch-Clamp Recording

The slices were removed from the holding chamber, placed in a recording chamber, and continuously perfused with oxygenated (95% O_2_; 5% CO_2_) aCSF at a rate of 2 mL per min. Whole-cell voltage-clamp recordings from layer II/III pyramidal neurons of the ACC cg1 region were obtained under visual guidance using a 40X objective on a Zeiss Axioskop FS upright microscope. Recordings were made with electrodes (4–6 MΩ) fabricated using a horizontal puller (P1000; Sutter, Novato, CA, USA) and filled with an internal solution containing (in mM) 120 Cs-MeSO_3_, 5 NaCl, 1 MgCl_2_, 0.5 EGTA, 2 Mg-ATP, 0.1 Na_3_GTP, 10 HEPES, and 5 QX314 (pH adjusted to 7.3 with CsOH, ~290 mOsmol). The neurons were voltage-clamped at −60 mV for recording EPSCs and 0 mV for IPSC using an Axon 700B amplifier (Axon Instruments, Foster City, CA, USA), low-pass filtered at 1 kHz, and digitized at 10 kHz with Clamplex (version 10.6; Molecular Devices, San Jose, CA, USA). The evoked EPSCs (eEPSCs) were stimulated by placing a tungsten bipolar stimulating electrode (Microprobes, Gaithersburg, MD, USA) in deep layers of the ACC. GABA_A_ (γ-aminobutyric acid type A)-receptor–mediated inhibitory synaptic currents were recorded in the presence of cyanquixaline (CNQX, 20 mM) and 2-amino-5-phosphonovaleric acid (APV, 50 mM) to block AMPA and NMDA receptors, respectively. The GABA_B_ receptor blocker, CGP-55845 (3 mM), was added to the bath solution for these experiments. For paired-pulse ratio (PPR) recordings, paired stimulation (50 ms apart) was performed every 30 s. Stable baseline recordings were obtained for 5 min, followed by the perfusion of pharmacological agents. Input and access resistance were monitored continuously throughout each experiment; experiments were terminated if these altered by >15%. Only recordings with stable holding current and series resistance maintained below 25 MΩ were considered for analysis. 

### 4.4. Synaptic Drive

Based on previous work, we calculated synaptic drive as the overall state of synaptic transmission in an individual neuron [35]. Spontaneous EPSCs and IPSCs were each recorded for 7 min before and after the application of DA (50 μM) for 10 min. The synaptic drive was calculated using the following formula:(1)$$Synaptic\;Drive=\frac{sEPSCFrequency*sEPSCAmplitude}{sIPSCFrequency*sIPSCAmplitude}$$

### 4.5. Complete Freund’s Adjuvant (CFA) Model of Inflammatory Pain

Complete Freund’s adjuvant (CFA; Sigma Aldrich, Oakville, ON, Canada) was injected subcutaneously in a volume of 20 μL into the plantar hind paws using a 100-μL microsyringe with a 30-gauge needle. ACC slices were prepared four days following CFA injection, as described above. The four-day post-CFA interval was selected because mice display persistent mechanical sensitivity without an observable anxiety phenotype [31].

### 4.6. Drugs and Solutions

Reagents used for aCSF, the internal pipette solution, dopamine hydrochloride, sulpiride, 6-cyano-7-nitroquinoxaline-2,3-dione (CNQX), APV, CGP-55845, and picrotoxin (PTX) were purchased from Sigma-Aldrich (Oakville, ON, Canada). All drugs were prepared fresh by dissolving them in distilled water. 

### 4.7. Data and Statistical Analysis

Data were collected and analyzed using pClamp 9.2 software (Molecular Devices, San Jose, CA, USA). One- or two-way analysis of variance (ANOVA), with or without repeated measures, was used as appropriate for experiments in which a washout phase after drug application was measured. Dunnett’s or Tukey’s HSD was used for post hoc analysis, where appropriate. For baseline analysis, the time between −4 min and +1 min was used for analysis, as stable baseline responses are expected to be present +1 min following drug application. We used paired *t*-test comparisons to determine whether baseline and drug effects differed significantly in the absence of the washout phase. For the *t*-test analysis of unpaired groups, Welch’s correction was used for unequal sample size between groups. * *p*  <  0.05 was considered as statistically significant.

## Figures and Tables

**Figure 1 ijms-24-11113-f001:**
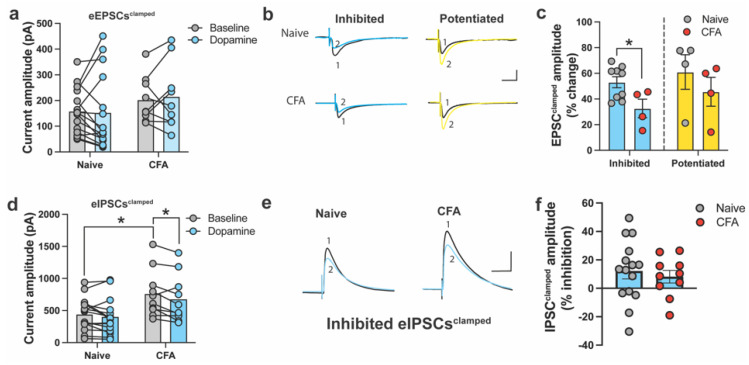
Dopamine (DA) and CFA treatment minimally affect excitatory (eEPSC^clamped^) and inhibitory eIPSC^clamped^) clamped currents when recorded without pharmacological isolation. (**a**) Bar graph showing average eEPSC^clamped^ amplitude (pA) before and after the application of dopamine (DA) in slices from naïve and CFA-treated mice (two-way ANOVA, main effect of CFA status: *F*_1,22_ = 1.33, *p* = 0.26; main effect of drug: *F*_1,22_ = 0.03, *p* = 0.8; drug × CFA interaction: *F*_1,22_ = 0.23, *p* = 0.63). (**b**) Representative traces for naïve and CFA-treated mice showing inhibited and potentiated eEPSCs^clamped^ before (1) and after (2) DA application. Scale bars = 40 ms, 200 pA. (**c**) Normalized comparison of percentage eEPSC^clamped^ inhibition and potentiation by DA in ACC neurons. In neurons classified as inhibited by DA, there is less overall inhibition by DA in CFA-treated neurons than in naïve neurons (*p* < 0.05). There is no difference in the overall inhibition by DA in neurons classified as potentiated (two-way ANOVA, main effect of CFA status: *F*_1,17_ = 4.479, *p* = 0.04; main effect of response: *F*_1,17_ = 1.524, *p* = 0.23; CFA × response interaction: *F*_1,17_ = 0.08, *p* = 0.77). (**d**) Bar graph showing average eIPSC^clamped^ amplitude (pA), before and after the dopamine (DA) application, in slices from naïve or CFA-treated mice. The overall eIPSC^clamped^ baseline responses of CFA-treated mice are larger than those of naïve mice (*p* < 0.05). DA inhibits eIPSC^clamped^ in CFA-treated mice, but not naïve mice (two-way ANOVA, main effect of CFA: *F*_1,23_ = 5.65, *p* = 0.03; main effect of DA: *F*_1,23_ = 8.390, *p* = 0.03; drug × CFA interaction: *F*_1,23_ = 0.98, *p* = 0.33). (**e**) Representative traces for naïve and CFA-treated mice showing inhibited eIPSCs^clamped^ before (1) and after (2) DA application. Scale bars = 40 ms, 200 pA. (**f**) Normalized comparison of percent of eIPSC^clamped^ inhibition by DA in ACC neurons. There is no difference in the percentage of eIPSC^clamped^ inhibition by DA (unpaired *t*-test, t_23_ = 0.51, *p* = 0.61). Data points below the *x*-axis represent neurons showing a potentiated response. In panels (**c**,**d**), Tukey’s HSD post hoc testing was used. * *p* < 0.05 compared with naïve or baseline responses. Bar graphs indicate mean ± SEM.

**Figure 2 ijms-24-11113-f002:**
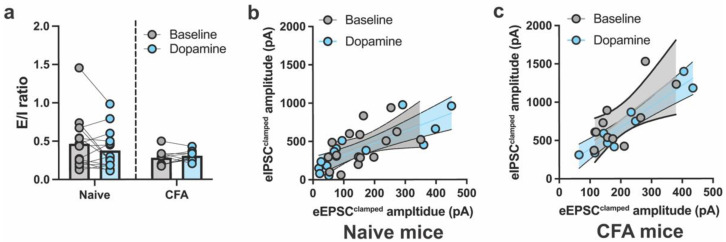
Dopamine modulation of excitatory/inhibitory balance in the ACC of naïve and CFA-treated mice. (**a**) DA does not significantly modulate the excitatory/inhibitory (E/I) ratio in naïve or CFA-treated mice (two-way ANOVA, main effect of CFA: *F*_1,23_ = 1.94, *p* = 0.17; main effect of DA: *F*_1,23_ = 0.69, *p* = 0.41; drug × CFA interaction: *F*_1,23_ = 2.26, *p* = 0.14). (**b**,**c**) Distribution of eIPSC^clamped^ and eEPSC^clamped^ amplitudes (pA) in ACC neurons prepared from naïve (**b**) and CFA-treated (**c**) mice. The slopes of the regression lines are similar before and after dopamine (DA) application in naïve (naïve baseline: *F*_1,13_ = 4.8, *p* = 0.04, *r*^2^ = 0.26; naïve DA: *F*_1,13_ = 27,30, *p* < 0.001, *r*^2^ = 0.67; difference between slopes: *F*_1,26_ = 0.019, *p* = 0.88) and CFA-treated mice (CFA baseline: *F*_1,8_ = 8.46, *p* = 0.02, *r*^2^ = 0.51; CFA DA: *F*_1,8_ = 55.78, *p* < 0.001, *r*^2^ = 0.87; difference between slopes: *F*_1,16_ = 0.013, *p* = 0.73). Bar graphs indicate mean ± SEM.

**Figure 3 ijms-24-11113-f003:**
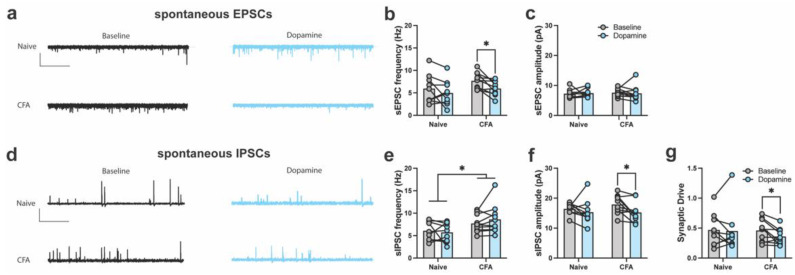
Dopamine modulates spontaneous excitatory and inhibitory currents in CFA-treated mice. (**a**) Sample traces of spontaneous (s)EPSCs in naïve (**top**) and CFA-treated (**bottom**) slices before and after application of dopamine (DA). Scale bars = 10 s, 200 pA. (**b**) DA reduces the frequency of sEPSCs in the ACC of CFA-treated mice (two-way ANOVA, main effect of CFA: *F*_1,17_ = 1.81, *p* = 0.19; main effect of DA: *F*_1,17_ = 8.77, *p* = 0.001; drug × CFA interaction: *F*_1,17_ = 0.63, *p* = 0.43). (**c**) DA did not change the amplitude of sEPSCs in the ACC of naïve or CFA-treated mice (two-way ANOVA, main effect of CFA: *F*_1,17_ = 0.05, *p* = 0.82; main effect of DA: *F*_1,17_ = 0.004, *p* = 0.94; drug × CFA interaction: *F*_1,17_ = 0.16, *p* = 0.68). (**d**) Sample traces of sIPSCs in naïve (**top**) and CFA-treated (**bottom**) slices before and after application of DA. Scale bars = 10 s, 20 pA. (**e**) CFA treatment increases the frequency of sIPSCs in naïve mice (two-way ANOVA, main effect of CFA: *F*_1,17_ = 5.03, *p* = 0.03; main effect of DA: *F*_1,17_ = 0.32, *p* = 0.57; drug × CFA interaction: *F*_1,17_ = 1.71, *p* = 0.21;). (**f**) DA application reduces the amplitude of sIPSCs in CFA-treated but not naïve mice (two-way ANOVA, main effect of CFA: *F*_1,17_ = 0.31, *p* = 0.59; main effect of DA: *F*_1,17_ = 5.7, *p* = 0.03; drug × CFA interaction: *F*_1,17_ = 1.01, *p* = 0.33). (**g**) DA decreases synaptic drive in CFA-treated but not naïve mice (two-way ANOVA, main effect of CFA: *F*_1,17_ = 0.19, *p* = 0.66; main effect of DA: *F*_1,17_ = 2.21, *p* = 0.15; drug × CFA interaction: *F*_1,17_ = 1.144, *p* = 0.29). A direct comparison of naïve and CFA-treated mice before and after DA application shows that synaptic drive is reduced by DA in CFA-treated but not naive mice (naïve: *t*_8_ = 0.23, *p* = 0.82; CFA-treated: *t*_9_ = 2.59, *p* = 0.02). In panels b, e, f, and g, Tukey’s HSD post hoc testing was used. * *p* < 0.05 compared with naïve or baseline responses. Bar graphs indicate mean ± SEM.

**Figure 4 ijms-24-11113-f004:**
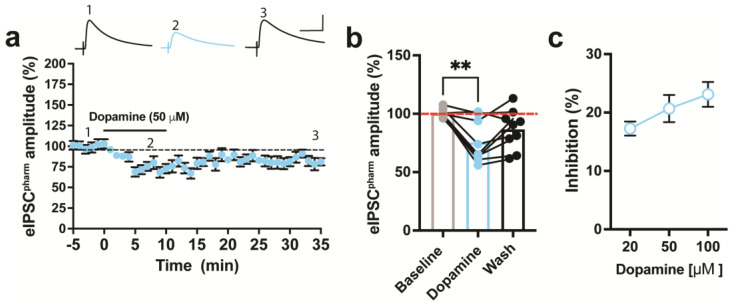
DA inhibits pharmacologically isolated evoked GABA_A_ receptor-mediated inhibitory currents (eIPSC^pharm^) from the ACC superficial layers. (**a**) **Top:** Sample traces of eIPSCs^pharm^ recorded at baseline (1), during dopamine (DA) application (2), and after washing (3). Scale bars = 40 ms, 200 pA. **Bottom:** Normalized data showing that the application of DA 50 μM inhibits eIPSCs^pharm^, which return towards baseline with subsequent washout of DA. (**b**) Inhibition of eIPSC^pharm^ by DA (50 μM), which returns to baseline (red dashed line) following washout (one-way repeated measure ANOVA, *F*_2,15_ = 12.28, *p* > 0.001; DA = 75.52% ± 5.4% of baseline, wash = 86.7% ± 5.37% of baseline). The comparison between baseline and wash is not statistically significant (*p* = 0.12), but the comparison between DA and wash is approaching significance (*p* = 0.069). (**c**) DA inhibition of eIPSCs^pharm^ does not differ at 20 μM (17.3 ± 1.2% of baseline), 50 μM (20.7 ± 12.3% of baseline), or 100 μM (23.1 ± 2.1% of baseline) (one-way ANOVA, *F*_2,16_ = 0.84, *p* = 0.4489). Symbols and bars indicate mean ± SEM. ** *p* < 0.01 compared to baseline using Dunnett’s case comparison.

**Figure 5 ijms-24-11113-f005:**
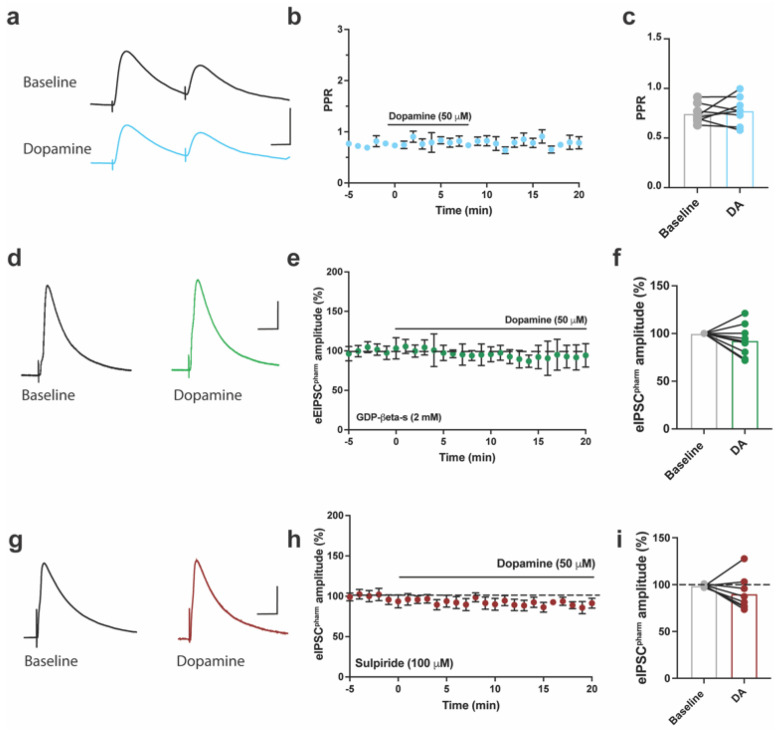
DA inhibits eIPSCs^pharm^ via postsynaptic GPCR−coupled D2 receptors. (**a**) Sample traces of GABA_A_-mediated eIPSCs^pharm^ from ACC superficial layers before and after DA application. Scale bars = 40 ms, 200 pA. (**b**) The application of DA (50 µM) did not change the paired–pulse ratio (PPR) of eIPSCs^pharm^. (**c**) No difference was noted in PPR at baseline and following DA application (paired *t*-test, *t*_8_ = 0.63, *p* = 0.54). (**d**) Sample traces of GABA_A_-mediated eIPSCs from ACC superficial layers were recorded with GDP-β-S (2 mM) in the recording pipette (40 ms, 200 pA). (**e**) Application of DA did not influence eIPSCs^pharm^ amplitude with GDP-β-S 2 mM in the recording pipette. (**f**) No difference was found in eIPSCs^pharm^ amplitude at baseline and following DA application with GDP-β-S 2 mM in the recording pipette (paired *t*-test, *t*_9_ = 1.53, *p* = 0.16). (**g**) Sample traces of GABA_A_-mediated eIPSCs^pharm^ from ACC superficial layers were recorded with a D2 receptor antagonist, sulpiride (40 ms, 200 pA). (**h**) Application of DA did not influence eIPSC^pharm^ amplitude in the presence of the sulpiride (100 mM). (**i**) No difference was found in eIPSCs^pharm^ at baseline and following application of DA with sulpiride (paired *t*-test, *t*_7_ = 1.336, *p* = 0.22. Symbols and bars indicate mean ± SEM.

## Data Availability

The datasets used and analyzed during the current study are available from the corresponding author upon reasonable request and with permission from the University of Toronto.

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
