# Peer review of "Influence of Inflammatory Pain and Dopamine on Synaptic Transmission in the Mouse ACC"

_ijms, 2023, doi:10.3390/ijms241311113_

Round 1
Reviewer 1 Report
Please also mention where the abbreviation CFA (complete Freund’s adjuvant ) comes from in order not to create confusion.
The four-day post-CFA interval was selected compared to a previous experiment, but it was also important to carry out a behavioral test to confirm the inflammatory process of the mice used in the experiment.
Author Response
We thank the reviewer for providing these suggestions. Below is our point-by-point response. Changes to the manuscript are also provided in red font.
Point 1. Please also mention where the abbreviation CFA (complete Freund’s adjuvant ) comes from in order not to create confusion.
Response. We have defined CFA in the abstract and at the end of the introduction so that there is no confusion regarding its terminology. We also included a brief description of CFA at the end of the introduction.
Point 2. The four-day post-CFA interval was selected compared to a previous experiment, but it was also important to carry out a behavioural test to confirm the inflammatory process of the mice used in the experiment.
Response. The mice used in the current paper were not checked for allodynia following CFA injection. We did not feel that it was necessary, given that CFA hind paw injection produces robust mechanical sensitivity that lasts upwards of 7 days. We have other data (along with our previously published work) in support of this claim, but we also do not feel that it is necessary to include them in the current paper. However, we now mention at the end of the discussion that CFA-induced allodynia was not verified, and this could be a contributing factor to the heterogeneity we observe in ACC electrophysiological responses.
Reviewer 2 Report
This paper investigates the effects of DA on synaptic transmission in the ACC of mice, focusing on inhibitory synaptic currents in the ACC in the context of inflammatory pain. The introduction to the work is sufficient, and the methods are well documented. The flow of the text in the article is very nice. However, I have certain comments, as listed below:
-
In Figure 1c, the p-value might not provide conclusive proof due to the extremely low number of samples in this group. To obtain more reliable and robust results, it is recommended to increase the sample size. Alternatively, a discussion can be included about the same.
-
It is essential to determine if there is a significant difference between the baseline and wash groups in Figure 4B. This is crucial for understanding the reversibility of inhibition observed in the study.
-
In addition to summarizing the findings and presenting contextual references and speculative accounts, it is crucial to include an overview of the significance of the obtained results in the discussion section.
Author Response
We thank the Reviewer for providing these suggestions. Our point-by-point response can be found below. Changes to the manuscript are also provided in red.
Point 1. In Figure 1c, the p-value might not provide conclusive proof due to the extremely low number of samples in this group. To obtain more reliable and robust results, it is recommended to increase the sample size. Alternatively, a discussion can be included about the same.
Response. This is a valid point. We have provided a discussion point stating that 'However, the overall power of these comparisons was low given that data were separated based on whether DA inhibited or potentiated the current. Thus, even though we report these findings as statistically significant, further validation and confirmation of these observations should be conducted using a larger sample size. This will improve the reliability and robustness of the results as a whole.
Point 2. It is essential to determine if there is a significant difference between the baseline and wash groups in Figure 4B. This is crucial for understanding the reversibility of inhibition observed in the study.
Response. To analyze these data, we used a one-way repeated measures ANOVA. In Figure 4B, we only reported the significant post-hoc comparison (Baseline vs DA). The comparison between the baseline and the wash comparison did not reach statistical significance, while the comparison between DA and wash was approaching statistical significance ( p = 0.069). We have now included this information and associated p-values in Figure 4 legend.
Point 3. In addition to summarizing the findings and presenting contextual references and speculative accounts, it is crucial to include an overview of the significance of the obtained results in the discussion section.
Response. We fully agree. We have now included an overview of the significance at the end of the discussion section. The last paragraph of the discussion now reads"Overall, our findings highlight the heterogeneity of ACC neuronal responses to DA, depending on pain state and presynaptic modulation. The implications of our results are twofold. Firstly, they emphasize the importance of considering the specific experimental conditions and the presence of presynaptic receptors when studying the effects of DA on synaptic currents in the ACC. This highlights the need for precise experimental design and methodology to accurately interpret DA modulation in neuronal circuits. Secondly, the results suggest that DA modulation in the ACC may play a role in pain-related mechanisms and potentially contribute to pain modulation. The observed reduction in presynaptic glutamate release in response to DA in CFA-treated mice and the heterogeneity of ACC neuronal responses imply a complex interplay between DA, pain processing, and synaptic transmission. However, we would like to note that we did not confirm CFA-induced mechanical sensitivity in the mice used for electrophysiological recordings, which may have played a role in the overall heterogeneity of ACC synaptic responses. Future studies are necessary to address the role of ACC DA signaling in pain processing in conditions with abnormal DA transmission."